# The Effects of Processing Parameters during the Wire Arc Additive Manufacturing of 308L Stainless Steel on the Formation of a Thin-Walled Structure

**DOI:** 10.3390/ma17061337

**Published:** 2024-03-14

**Authors:** Bang Liu, Jun Lan, Hongqiang Liu, Xinya Chen, Xin Zhang, Zhengyi Jiang, Jian Han

**Affiliations:** 1School of Materials Science and Engineering, Tianjin University of Technology, Tianjin 300384, China; bangliu1998@163.com (B.L.); lj526586@163.com (J.L.); cxyhxsx@163.com (X.C.); 2Technology Research Institute, HBIS Group, Shijiazhuang 050023, China; ajwei@163.com; 3School of Automobile Engineer, Changshu Institute of Technology, Suzhou 215556, China; zgxin99@163.com; 4School of Mechanical, Materials, Mechatronic and Biomedical Engineering, University of Wollongong, Wollongong, NSW 2522, Australia; jiang@uow.edu.au

**Keywords:** wire arc additive manufacturing, 308L stainless steel, processing parameters, dimensional accuracy

## Abstract

Wire arc additive manufacturing (WAAM) excels in producing medium to large components with a high deposition rate. Process optimization is crucial for uniform, defect-free components. This research employs orthogonal experimental design and response surface methodology (RSM) to control TIG WAAM-ed 308L stainless steel components. Varied parameters, including tungsten electrode angle, welding current, and speed, target weld bead attributes. Analysis of variance (ANOVA) evaluates multi-processing parameter influence on weld bead formation. Comparison with experimental results confirms accurate modeling of the relationship between parameters and bead attributes. The study optimizes process parameters and swing to enhance dimensional accuracy in single-layer and multi-layer components, improving precision, quality, and accuracy in thin-walled structures.

## 1. Introduction

308L stainless steel was widely applied in various fields due to its excellent corrosion resistance, high-temperature resistance, and weldability [1,2,3,4]. WAAM emerged as a promising method for producing large, complex metal structures, offering advantages such as high process rates, material efficiency, low cost, and reduced risks of pollution and porosity. As a result, it was found that the applications in diversified industries, including automotive, aerospace, and biomedical sectors. These advantages attracted researchers and industries to further develop WAAM, aiming at enhancing its process performance, process capabilities, and applicability in various fields [5,6,7,8,9]. The factors influencing the shaping process in WAAM include welding current, welding of shielding gas, nozzle-to-substrate distance, and wire feed angle [10]. Due to the numerous factors affecting the shaping process and their interdependencies, optimizing the process becomes quite challenging. In the latest research, Benakis et al. [11] summarized the weld bead geometric features in WAAM. The total height of the weld bead was defined by the reinforcement height (r) and the penetration depth (p). In WAAM, it was possible to vary the ratio of weld bead height to width (w) to achieve different geometric shapes for the root and cover passes. Therefore, it was necessary to study the variations in weld bead size measurements using different combinations of process parameters. Erfanmanesh et al. [12] adopted an empirical–statistical approach to optimize the laser cladding of WC-12Co powder on the AISI 321 steel substrate. They utilized regression analysis (RA) to predict and analyze the correlation between the key process parameters (laser power, scanning speed, and powder feed rate) and the geometric characteristics of the single cladding layer (height, width, dilution rate, and wetting angle). Eventually, they obtained optimal process parameters for the laser cladding of WC-12Co powder on the AISI 321 stainless steel surface. Yadav et al. [13] established mathematical models to study the influence of welding speed, wire feed speed, and welding voltage on weld bead geometry and weld dilution. They employed a design of a central composite rotational factor, conducting statistical analysis with three factors at five levels. Response surface methodology (RSM) was used to analyze the results. Variance analysis was performed to examine the adequacy of the models and parameters for welding process optimization. U. de Oliveira et al. [14] conducted a study from both theoretical and experimental perspectives on the process of depositing thick nickel-based coatings on a steel substrate using a Nd:YAG 2 kW continuous-wave laser. Chen et al. [15], via the analysis of waveforms and metal transfer images simultaneously detected under different welding conditions, established the distribution of CMT cycles and short-circuit times. The results indicated that the deposition rate, average wire feed speed, and droplet size increased with the increase in either the voltage or the duration of voltage rise, even when the preset wire feed speed remained constant. With the increase in voltage or voltage rise duration, the weld bead width and penetration depth increased significantly, while the strengthening effect on carbon steel remained relatively small. Sun et al. [16] focused on the influence of laser power (400–600 W), scanning speed (500–700 mm/min), and powder feed rate (30–60 rev/min) on the shape factor and the cladding-bead geometry (layer width, layer height, and molten depth) with regard to injecting Ti6Al4V powder on Ti6Al4V substrate. Therefore, emphasis should be on dimensional precision and surface finish in WAAM [17,18,19].

In this study, we focused on three key process parameters: tungsten electrode angle, welding current, and welding speed to describe the process. Although there were already numerous physical models for the WAAM process [20,21], empirical–statistical models helped avoid the complexity of analyzing physical phenomena. Therefore, empirical–statistical research was beneficial for modeling the WAAM process [12]. To reduce the number of experiments, an orthogonal experimental method was adopted for parameter optimization [22,23,24]. This method employed a central composite design to identify the effects of defined input variables, such as tungsten electrode angle, welding current, and welding speed. Ultimately, it established the relationships between these variables and the corresponding output responses. Firstly, a design matrix was generated based on the three process parameters, and these variables varied at five levels. Additionally, ANOVA was employed to determine the significance of individual input parameters on independent output responses. The weld bead height, bead depth, bead width, wetting angle, and dilution rate were measured using an optical microscope as responses, and further analysis was conducted using Design-Expert 10 statistical software. The suitable process parameters were ultimately determined. Additionally, via further optimization of the process parameters and investigating the influence of swing on dimensional accuracy in single-pass, multi-layer fabrication, rational solutions were provided for the issues encountered in single-pass, multi-layer AM.

## 2. Materials and Methods

### 2.1. Experimental Setup

The additive manufacturing experiment conducted in this study used a tungsten inert gas (TIG) welding system with inert gas protection. Once the experimental parameters were set, the process allowed for the realization of fully automated welding. The physical representation of the entire WAAM system (Wuxi Gutu Welding Equipment Co., Ltd., Wuxi, China) is shown in Figure 1a. The thin-walled part with geometrical features is shown in Figure 1b [25,26]. The platform primarily consisted of a direct current welding power source, control cabinet, control panel, argon gas protection device with an accompanying hood, work platform, wire feeding mechanism, and welding torch.

### 2.2. Materials

In this experiment, the welding wire used was ER-308L with a diameter of 1.2 mm. The substrate employed was a 304 stainless steel substrate with dimensions of 250 × 150 × 8 mm. Before the experiment, the substrate surface was polished to a bright finish using 120# sandpaper and cleaned with alcohol. The chemical compositions of the 308L welding wire and the 304 substrate are shown in Table 1.

### 2.3. Design of Experiments (DOE)

RSM can be used to model, optimize, and analyze problems where the interested response is influenced by multiple variables. It fits a multivariate quadratic regression equation to describe the functional relationship between the factors and response values, ultimately aiming to optimize the response. In the context of WAAM, the main process parameters include tungsten electrode angle, welding current, and welding speed. Since their impact on weld bead formation is not independent, based on the principles of central composite design (CCD), the experiment selected these three process parameters as variable factors, each with five levels. The response surface experiment was designed with three factors and five levels, using bead height, bead depth, bead width, wetting angle, and dilution rate as response values.

### 2.4. Metallographic Preparation and Observation

In this study, the polished 308L metallographic specimens were subjected to etching using aqua regia (HCl:HNO_3_ = 3:1) for 5 s. Nitric acid can oxidize the stainless steel surface with the formation of an oxide layer, and the HCl content of aqua regia can dissolve that. So, the surface is cleaned and, due to short contact time, is passivated. Observation of the weld bead section shape and the thin-wall wall section was conducted using an OLYMPUS GX51(Olympus Corporation, Tokyo, Japan) metallurgical optical microscope (OM). ImageJ 1.51 software was utilized to measure the geometric characteristics of the weld bead, including the bead height, bead depth, bead width, wetting angle, and dilution rate.

## 3. Results and Discussion

### 3.1. Single-Pass Single-Layer Welding Seam Formation Test

In this experiment, the WAAM process was employed to conduct the single-pass, single-layer wall-forming test of 308L stainless steel. Therefore, in this experiment, a large number of single-pass, single-layer tests were conducted, focusing on three factors: tungsten electrode angle, welding current, and welding speed. These experiments aimed to determine the favorable range of process parameters for single-pass weld bead formation and provide a basis for subsequent additive manufacturing. Currently, there is a relatively limited amount of research regarding 308L stainless steel WAAM; thus, the results of these experiments will offer valuable references and guidance in this field. Via these single-pass, single-layer tests, we identified the optimal range of process parameters, laying a more reliable foundation for single-pass, multi-layer wall formation and future batch production of large-sized components via WAAM. Other fixed process parameters are listed in Table 2.

In the single-pass, single-layer weld bead experiments, the process parameters that were varied included tungsten electrode angle, welding current, and welding speed. As their effects on weld bead formation were not independent, based on the principles of CCD, we selected these three welding parameters as variable factors. The response surface experiment involved using bead height, bead width, bead depth, wetting angle, and dilution rate as response values. A three-factor, five-level response surface experiment was designed, with each factor having five levels. The variation ranges for each factor are shown in Table 3, and all subsequent experimental combinations were based on these ranges.

Using Design-Expert 10 software, a CCD was employed to determine the experimental plan for investigating the effects of tungsten electrode angle, welding current, and welding speed on bead height, bead depth, bead width, wetting angle, and dilution rate. A total of 25 sets of process parameter combinations were generated. The results and corresponding response values are shown in Table 4.

Figure 2 displays the variations in characteristic parameters of 308L thin-wall single-pass welds. The weld bead height, depth, width, and dilution rate of 308L stainless steel exhibited typical normal distributions, while the wetting angle followed a bimodal distribution pattern. This was because the wetting angle was significantly negatively correlated with dilution rate and height but unrelated to width. In practical applications, suitable process parameters can be chosen based on the characteristic parameters of 308L stainless steel, and a process parameter database can be established based on different characteristic parameter results, as shown in Figure 2f. For easy reading of the axes, they are shown in different colors on different mapping surfaces respectively. Since different materials have varying characteristic parameters, their process parameters also vary.

Figure 3 shows the macroscopic morphology of the 25 different combinations of single-pass, single-layer weld beads. Most of the specimens exhibited slight oxidation on the surface of the weld bead and in the heat-affected zone of the base metal. During the process, if the substrate temperature was relatively low before the first process layer transitioned to the base metal, the molten metal might not spread well on the substrate surface due to insufficient surface tension. Moreover, the welding speed and wire feeding speed are mismatched, resulting in intermittent point-like process layers and the inability to form a continuous process layer [27]. This morphology is shown in red in Figure 3 (samples numbered 6, 7, 11, 16, 21, 22, and 23). In such cases, due to the significant and discontinuous height variation of the weld bead, the beads were considered unsuitable for AM applications. Three metallographic samples were taken at different locations within each weld bead, and Figure 4 shows the corresponding cross-sectional metallographic images.

In Figure 5a, a cross section of a general weld bead geometry with high penetration is presented [11]. While this geometry is ideal for the root pass on the substrate, it is not beneficial for the bead height, as high penetration depth liquefies the previous passes and reduces the existing bead height by widening the melt pool. By reducing the bead depth and increasing the bead height, more flattened beads are formed (Figure 5b). An additional increase in the bead height results in more “bumpy” beads (Figure 5c) where the bead height exceeds the bead depth of penetration. Finally, by reducing the bead width while increasing the bead height, a more rounded ball-shaped bead is formed (Figure 5d). The calculation method for dilution rate is shown in Figure 5e [28].

### 3.2. Influence of Process Parameters on Single-Pass Single-Layer Formation

#### 3.2.1. Effect of Process Parameters on Bead Height

The significance level of the input parameters was determined using the ANOVA. In order to obtain the best mathematical models, quadratic, linear, and two-factor interaction models were considered. For a selected response, the regression calculation was performed, and the model’s fitness was determined. The statistics, such as the probability of errors (*p*-value), lack of fit, and R-squared values for comparing the models were obtained to find the total deviation of the variables for the individual model. The R-squared values for the quadratic polynomial were 0.9598 (bead height), 0.7342 (bead depth), 0.8433 (bead width), 0.8630 (wetting angle), and 0.8196 (dilution rate), and show the minimum deviation of the variables. Therefore, the quadratic model was opted to analyze the variance between the input variables and output responses [12,20,21]. The second-order quadratic polynomial was used to represent the optimum response surface shown in Equation (1).
(1)$$Y=b_{0}+b_{1}X_{1}+b_{2}X_{2}+b_{3}X_{3}+b_{4}X_{1}^{2}+b_{5}X_{2}^{2}+b_{6}X_{3}^{2}+b_{7}X_{1}X_{2}+b_{8}X_{1}X_{3}+b_{9}X_{2}X_{3}$$
where *Y* is the estimated response, *b*_0_ is the constant, and *b*_1_ to *b*_9_ represent the linear and interactive coefficients. The terms *X*_1_, *X*_2_, *X*_3_, *X*_1_^2^, *X*_2_^2^, *X*_3_^2^, *X*_1_*X*_2_, *X*_1_*X*_3_, and *X*_2_*X*_3_ correspond to the independent factors (process parameters). The correlation coefficients (*R*^2^), adjusted *R*^2^, and predicted *R*^2^ values were determined using Equations (2)–(4), respectively [20].
(2)$$R^{2}=1-\left[SS_{\mathrm{residual}}/\left(SS_{\mathrm{residual}}+SS_{\mathrm{model}\;}\right)\right]$$
(3)$$\begin{array}{l} {adjusted\;R}^{2}=1-[\left(SS_{\mathrm{residual}}/df_{\mathrm{residual}}\right)/\left(SS_{\mathrm{residual}}+SS_{\mathrm{model}}/df_{\mathrm{residual}}+df_{\mathrm{model}}\right)] \end{array}$$
(4)$${predicted\;R}^{2}=1-\left[{\left(\sum_{i=1}^{n}e_{i}/\left(1-h_{ii}\right)\right)}^{2}/(SS_{\mathrm{residual}}+SS_{\mathrm{model}})\right]$$

Equations (5)–(9) display the coded equations of mathematical models developed in software, where *A*, *B*, and *C* are tungsten electrode angle, welding current, and welding speed, respectively.
(5)$$\begin{array}{c} Bead\;height=3076.94+66.19\times A-980.74\times B-297.35\times C-64\times AB-0.1038\times AC\\ -\;222.75\times BC+487.65\times A^{2}+207.05\times B^{2}+144.81\times C^{2} \end{array}$$
(6)Bead depth=662.83+66.08×A+347.01×B−162.74×C
(7)Bead width=5664.48+391.52×A+2274.41×B−170.53×C
(8)wetting angle=93.36−1.65×A−31.74×B−5.97×C
(9)Dilution rate=12.51+1.91×A+11.12×B−1.16×C

Figure 6 depicts the influence of various single factors on the bead height, along with a comparison between the predicted values and the response values. It is evident that welding current has the most significant impact on bead height, followed by welding speed. Moreover, bead height exhibits a trend of initially decreasing and then increasing with the increase in tungsten electrode angle, with the least significant effect observed at a tungsten electrode angle of 35°, where the black solid line in Figure 6a–c shows the actual results and the blue dashed line shows the error tolerance range. This is due to the inherent characteristics of TIG welding, which involve a substantial heat input, causing variations in the heat input to fluctuate over a wide range due to changes in welding current during the transition of the molten droplet. Such substantial fluctuations in heat input have a significant impact on the weld penetration depth. Concerning welding speed, an increase in welding speed reduces the amount of wire process per unit time on the weld seam, consequently decreasing the penetration depth. Excessive welding speed results in reduced linear energy input, causing the arc to not sufficiently melt the base metal and wire, leading to defects such as poor weld seam formation and undercut. On the other hand, a reduction in welding speed results in an increase in the amount of heat received per unit area, leading to an augmented depth of fusion. The tungsten electrode angle, tapered at its end, affects both the weld bead width and depth under the same welding current. A smaller θ angle induces arc column spreading, resulting in reduced weld penetration depth but increased bead width. As the angle increases, arc column spreading diminishes, leading to increased penetration depth but reduced bead width. Furthermore, these effects become more pronounced with higher welding current. To validate the accuracy of the regression equation fitting, a comparison between the predicted and actual values of melting height was performed. It can be observed that the actual values are closely distributed near the diagonal line, indicating that the regression equation effectively reflects the relationships between various single factors and melting height. In response to surface methodology, the F-value (resulting from the test for homogeneity of variances) is used to assess the significance of the established mathematical model. Typically, an F-value greater than 1 and a larger F-value at the chosen significance level suggest that the model is significant. The higher the F-value, the greater the significance of the model. Table 5 was analyzed using response surface methodology, yielding an F-value of 39.75, indicating the model’s significance. The F-value has only a 0.01% probability of increasing due to random interference. When the Prob > F value is less than 0.05, the respective parameter is considered significant; when the Prob > F value is less than 0.01, it is considered highly significant. From the data in Table 5, it is evident that the Prob > F value is less than 0.05, signifying the significance of this model.

The impact of correlated variations in input variables and responses was envisaged from the response surface analysis, as illustrated in Figure 7. The interaction effects of the parameters can also be seen from the 3D surface plots. Furthermore, by holding the third variable at its central level, interactions between the various parameters can be examined. Figure 7 displays the pairwise interactions between tungsten electrode angle, welding current, and welding speed. The interaction between the two variables significantly influences the bead height, with bead height increasing as the welding current decreases and the tungsten electrode angle increases. At low welding currents and relatively low welding speeds, bead height experiences enhancement. With low welding current and low welding speed, a minimal amount of heat energy is imparted to the workpiece, ultimately leading to increased bead height.

#### 3.2.2. Effect of Process Parameters on Bead Depth

Figure 8 presents an analysis of the influence of different single factors on bead depth, along with a comparison between predicted values and actual responses. It is evident that welding current has the most significant impact on bead depth, followed by welding speed, while tungsten electrode angle has the least influence on bead depth. By examining the relationship between predicted and actual bead depth values, the accuracy of the regression equation fitting can be assessed. It can be observed that actual values are closely distributed near the diagonal line, indicating that the regression equation effectively captures the relationship between welding parameters and weld penetration depth. Table 6, analyzed via RSM, demonstrates a model F-value of 19.33, indicating the model’s significance.

#### 3.2.3. Effect of Process Parameters on Bead Width

Figure 9 illustrates the impact of various single factors on bead width, along with a comparison between predicted values and actual responses. It is evident that welding current has the most significant influence on bead width, while welding speed and tungsten electrode angle have a relatively minor effect. By examining the relationship between predicted and actual bead width values, the accuracy of the regression equation fitting can be assessed. Actual values are observed to be closely distributed near the diagonal line, indicating that the regression equation effectively captures the relationship between process parameters and weld width. Table 7, analyzed via RSM, reveals a model F-value of 37.68, signifying the model’s significance.

#### 3.2.4. Effect of Process Parameters on Wetting Angle

Figure 10 depicts the influence of various single factors on the wetting angle, along with a comparison between predicted values and actual responses. It is evident that welding current has the most significant impact on wetting angle, while welding speed and tungsten electrode angle have a relatively minor effect. By examining the relationship between predicted and actual wetting angle values, the accuracy of the regression equation fitting can be assessed. Actual values are observed to be closely distributed near the diagonal line, indicating that the regression equation effectively captures the relationship between welding parameters and wetting angle. Table 8, analyzed via RSM, reveals a model F-value of 44.08, indicating the model’s significance.

#### 3.2.5. Effect of Process Parameters on Dilution Rate

Figure 11 illustrates the impact of various single factors on the dilution ratio, along with a comparison between predicted values and actual responses. It is evident that welding current has the most significant influence on the dilution ratio, while welding speed and tungsten electrode angle have a relatively minor effect. By examining the relationship between predicted and actual dilution ratio values, the accuracy of the regression equation fitting can be assessed. Actual values are observed to be closely distributed near the diagonal line, indicating that the regression equation effectively captures the relationship between welding parameters and dilution ratio. Table 9, analyzed via RSM, reveals a model F-value of 31.80, signifying the model’s significance.

After establishing the predictive model, numerical optimization using the desirability function analysis was employed to optimize the response variables. The objective of optimization was to identify the optimal settings that ensure the production of high-efficiency and high-quality end products in practical manufacturing processes. Determined best response value result intervals: bead height (2400–2800 μm), bead depth (350–1250 μm), bead width (3000–9000 μm), wetting angle (55–95°), dilution rate (4–28%). The selected parameters for subsequent shaping experiments were as follows: tungsten electrode angle of 35°, welding current of 135 A, and welding speed of 100 mm/min.

### 3.3. Influence of Process Parameters on Accuracy of Single-Pass Multi-Layer Formation

#### 3.3.1. Influence of Parameter Optimization on Accuracy of Single-Pass Multi-Layer Formation

To ensure the suitability of the selected welding parameters post-screening, further optimization was conducted by employing the method of controlled variables. The optimization objective was to find the optimal parameter combination to either maximize or minimize a specific performance metric. Three new parameter combinations were chosen for experimentation: (a) tungsten electrode angle of 35°, welding current of 135 A, and welding speed of 100 mm/min; (b) tungsten electrode angle of 35°, welding current of 145 A, and welding speed of 100 mm/min; and (c) tungsten electrode angle of 35°, welding current of 135 A, and welding speed of 120 mm/min. Experimental trials were conducted with these optimized welding parameter combinations, depositing single-pass multi-layered single-wall structures. The macroscopic appearances of the resulting structures are depicted in Figure 12a–c. We began by observing and describing the actual appearance of the weld seam. The weld seam surface exhibited clear signs of oxidation, appearing as a deep brown color rather than the expected silver-gray or bright copper color. Irregular oxidation spots and textures were also present on the surface. The weld seam exhibits slight unevenness. To address this issue, we subsequently introduced oscillation while keeping the parameters constant.

#### 3.3.2. Influence of Swing on Accuracy of Single-Pass Multi-Layer Formation

Figure 12d–f present the macroscopic appearances of single-pass weld seams after applying an 8 mm swing. In comparison to the weld seam formation without swing, the results show improved formation quality with a lack of surface defects and a lower degree of oxidation. Swing helps improve the uniformity and coverage of the weld seam. It ensures that the wire is fully melted and bonded, reducing the occurrence of pores and uneven melting. This resulted in a more stable and reliable welding quality. Figure 13 depicts the scale patterns formed during swing and non-swing processes. In the rapid solidification of metals, scale patterns result from uneven cooling of the molten pool and the interaction of fluid dynamics. These patterns exhibit shapes correlated with the swing path.

Figure 14 presents cross-sectional micrographs after non-swing and swing processes, demonstrating well-formed bonds without any defects, such as lack of fusion or porosity across all process parameters. Table 10 displays the results of the response values after parameter optimization and swing.

Figure 15 illustrates the single-pass multi-layer thin-wall structures after parameter optimization and swing. During the process, in the initial layer transitioning to the substrate, the lower substrate temperature may lead to insufficient surface tension of the molten metal, resulting in an intermittent process and the inability to form a continuous layer. Additionally, the substrate metal must exhibit good flatness to ensure the stability of the layer process. During the wall-building additive process, at the arc initiation point, the instantaneous formation of the arc causes an unstable melting process, leading to a relatively high process rate. Conversely, during arc termination, as the arc slowly extinguishes, the molten metal continues to spread in the direction of the layering process under the influence of arc forces. Consequently, there is a lower process rate at the termination point. During the process in the same direction, this variation accumulates repeatedly. Initially, the height at the starting position is higher than in the stable portion. As the height increases toward the end, molten metal continuously flows downward, ultimately resulting in a wall tilted from the horizontal plane, as depicted in Figure 15a–c. When an 8 mm swing is applied, the surface formation improves, and there is no significant collapse at the tail end, as shown in Figure 15d–f.

Surface roughness has a significant impact on the service life and reliability of metal parts [29]. The surface shape accuracy of components was crucial for their fit, assembly, and performance with other components. In the additive manufacturing process, temperature gradients led to residual stress and deformation, while the quality of connections between different layers directly impacted the strength and durability of the components. Figure 16 presents the side profile roughness of the single-pass multi-layer thin-wall structures after parameter optimization and swing. The selected region for measurement corresponds to the effective area outlined in Figure 15. Figure 17 displays the three-dimensional morphology of the selected yellow region in Figure 15. By combining the line roughness data from Figure 16 with the maximum height difference in the three-dimensional morphology of the effective area in Figure 17, it is evident that roughness is significantly reduced after the swing. Among the welding parameter combinations, the one with the least roughness occurs when utilizing a tungsten electrode angle of 35°, welding current of 135 A, welding speed of 120 mm/min, and an 8 mm swing. In this scenario, the roughness is only 64.046, representing an 89.58% reduction compared to the absence of swing. This greatly enhances the formation quality of the thin-wall structure. Roughness affects aspects such as part assembly, wear resistance, fatigue strength, and corrosion resistance and is one of the commonly used inspection parameters in actual production. The arithmetic mean deviation of the profile, Ra, is the average arithmetic deviation of the absolute distances from each point on the profile to the centerline within a certain measurement length l. It is represented by the following formula:(10)$$Ra=\frac{1}{l}\int_{0}^{l}|y|dx$$

The larger the numerical value, the rougher the surface of the part.

Figure 18 presents the cross-sectional morphology of the single-pass multi-layer thin-wall structures after parameter optimization and swing. Table 11 shows the calculated molding heights and molding efficiencies for the corresponding cross sections of Figure 18. Figure 18g illustrates the concept of formation efficiency, which can be quantified using the following formula [30,31]:(11)$$\eta =\frac{A}{S}$$

Here, region *A* corresponds to the area available for utilization within the cross-section, and region *S* corresponds to the total cross-sectional area. Figure 19 illustrates schematic diagrams of non-swing and swing portions. It is observed that a lower bead width-to-height ratio (i.e., the ratio of bead width to bead height) leads to higher surface waviness and lower process efficiency (Figure 19a), whereas a higher bead width-to-height ratio ensures closer adherence to the desired form and higher process efficiency (Figure 19b). Therefore, at a given wire feed rate, a higher bead length-to-diameter ratio results in better surface smoothness of the component and higher process efficiency.

Figure 20 illustrates a comparison of the formed heights and schematic representations of the formation efficiency for the single-pass multi-layer thin-wall structures after parameter optimization and swing. The application of swing results in an increase in the thickness of the thin-wall structure, consequently leading to a slight reduction in the formed height. The wall can distribute the material more evenly and reduce gaps between layers. Therefore, the formation efficiency is significantly improved compared to the non-swing scenario, with enhancements of 24%, 24%, and 31%, reaching 88%, 91%, and 94%, respectively. This substantial increase in efficiency greatly reduces material wastage during subsequent processing stages.

## 4. Conclusions

The macroscopic features, quality, and precision of the workpiece were closely related to the process parameters, such as tungsten electrode angle, welding current, and welding speed. For different geometric characteristics, the influence of these three process parameters on the geometric accuracy of the WAAM process was discussed. The process data and methodology provided by this paper might be helpful to the studies and explorations of the process of additive manufacturing. The model analysis and experimental results within the study indicate that concerning individual factors, welding current exerts the most significant influence on weld seam formation, while welding speed and tungsten electrode angle have relatively minor impacts. Additionally, there is a notable interaction effect between welding current and tungsten electrode angle on weld seam height. Following optimization using Design-Expert 10 software, we determined the best response value result intervals: bead height (2400–2800 μm), bead depth (350–1250 μm), bead width (3000–9000 μm), wetting angle (55–95°), and dilution rate (4–28%). Parameter optimization can further refine the selection of suitable forming parameter combinations, thereby controlling the precision of thin-wall structure formation. The swing process results in a faster cooling rate, leading to lower heat accumulation in the thin-wall structure compared to non-oscillated counterparts. It also enhances surface precision and improves formation efficiency. Ultimately, the optimal parameter combination for forming thin-wall structures consists of welding current: 135 A, welding speed: 120 mm/min, tungsten electrode angle: 35°, and an 8 mm swing.

## Figures and Tables

**Figure 1 materials-17-01337-f001:**
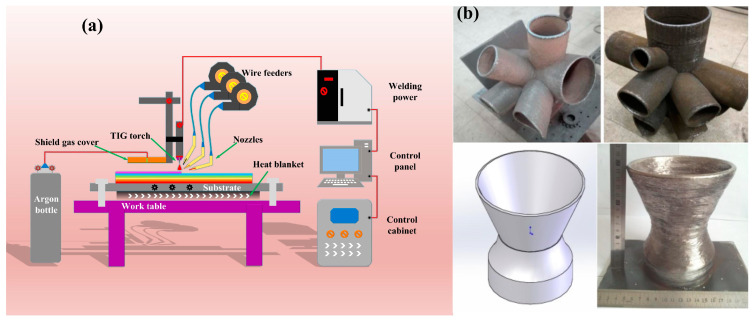
(**a**) Schematic diagram of the WAAM system; (**b**) a typical thin-walled part with geometrical features [25,26].

**Figure 2 materials-17-01337-f002:**
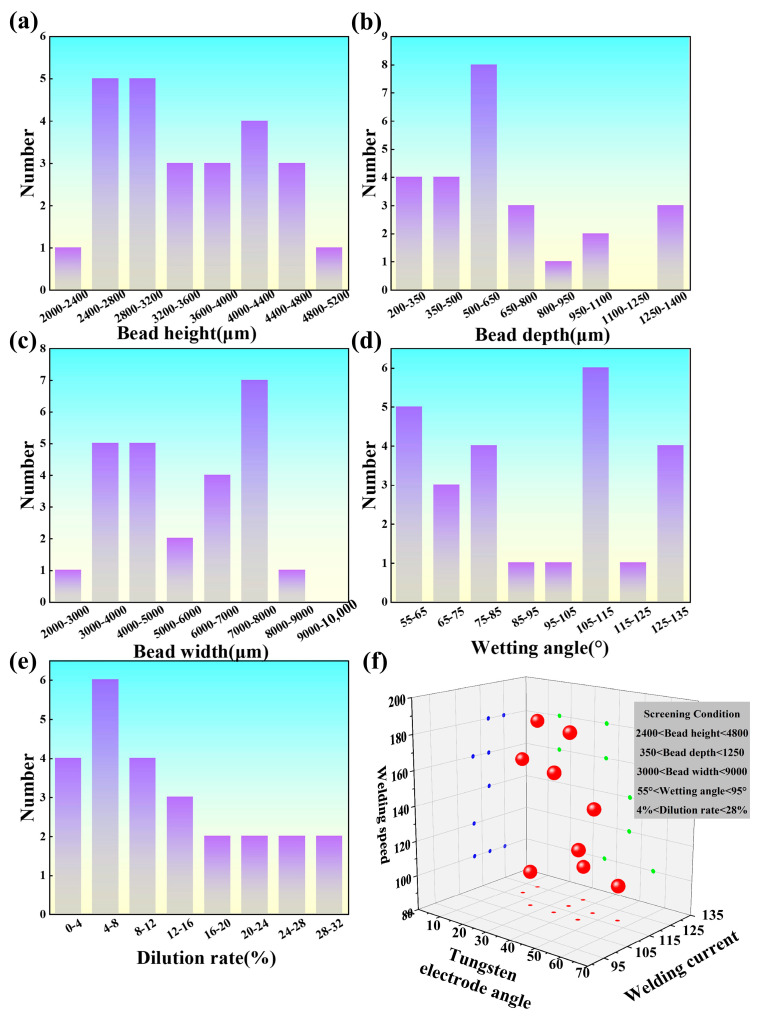
Variation rules of the characteristic parameters of the 308L thin-walled single-pass deposits: (**a**) bead height; (**b**) bead depth; (**c**) bead width; (**d**) wetting angle, (**e**) dilution rate; (**f**) WAAM process parameter databases (tungsten electrode angle, welding current, and welding speed) of the 308L according to the requirements.

**Figure 3 materials-17-01337-f003:**
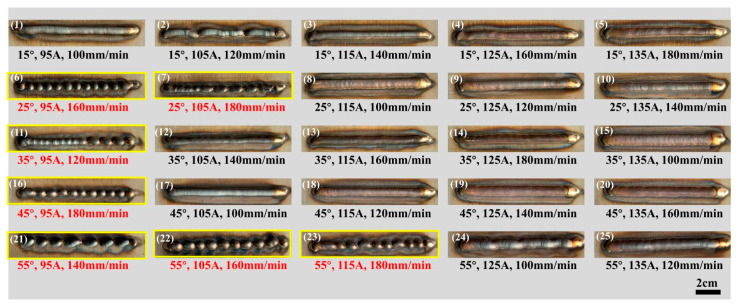
CCD-based design for single-pass, single-layer weld testing.

**Figure 4 materials-17-01337-f004:**
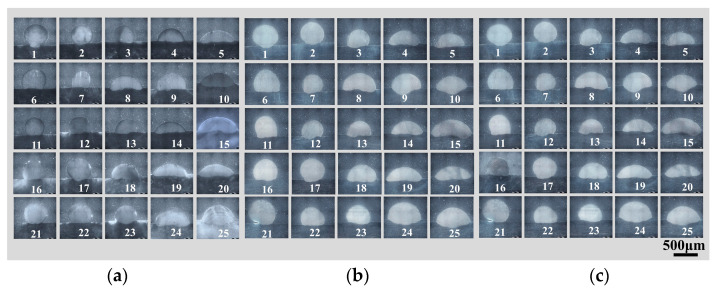
Single layer interface with different process parameters: (**a**) head, (**b**) middle, and (**c**) tail.

**Figure 5 materials-17-01337-f005:**
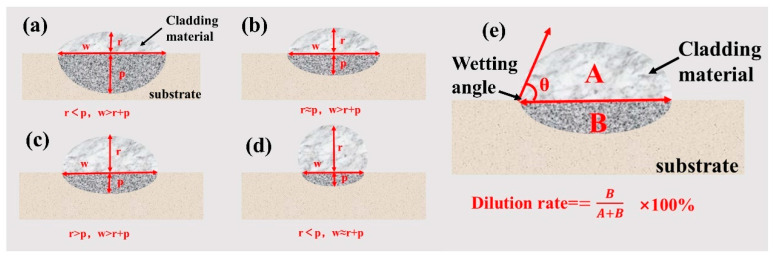
Schematic diagram of the geometric characteristics of a single-pass, single-layer weld: bead height (r), bead depth (p), and bead weight (w), (**a**) r < p, w > r + p, (**b**) r ≈ p, w > r + p, (**c**) r > p, w > r + p, (**d**) r < p, w ≈ r + p, (**e**) the calculation method for dilution rate.

**Figure 6 materials-17-01337-f006:**
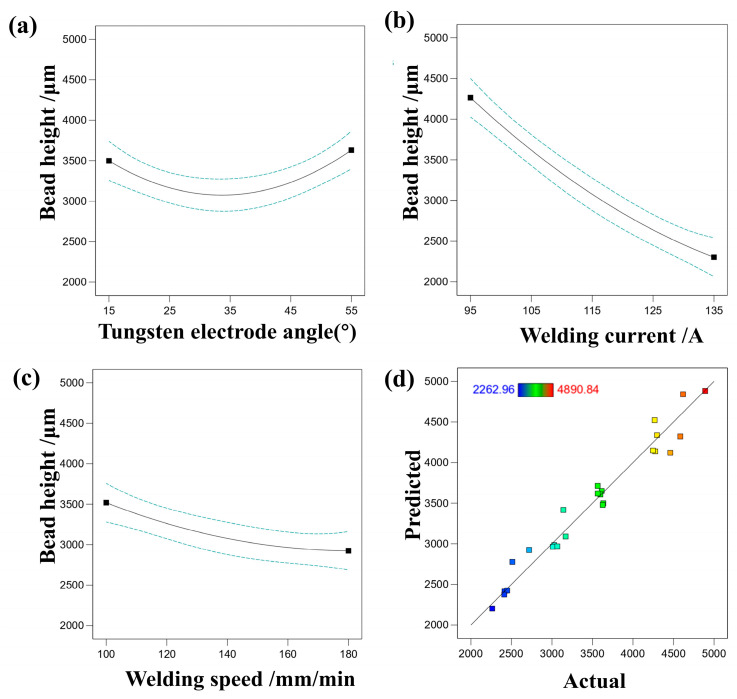
(**a**) The effect of tungsten electrode angle on bead height, (**b**) the effect of welding current on bead height, (**c**) the effect of welding speed on bead height, (**d**) comparison of predicted and actual values of bead height.

**Figure 7 materials-17-01337-f007:**
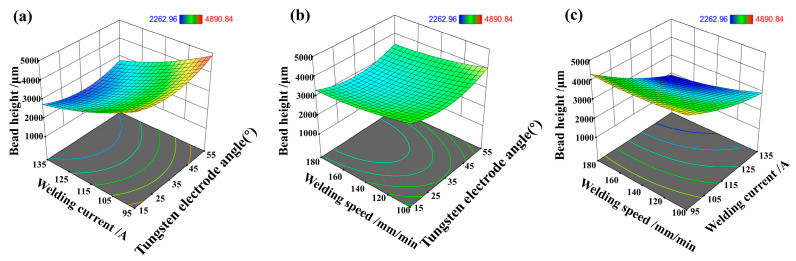
(**a**) The interaction effect of tungsten electrode angle and welding current on the melt height, (**b**) The interaction effect of tungsten electrode angle and welding speed on the melt height, (**c**) The interaction effect of welding speed angle and welding current on the melt height.

**Figure 8 materials-17-01337-f008:**
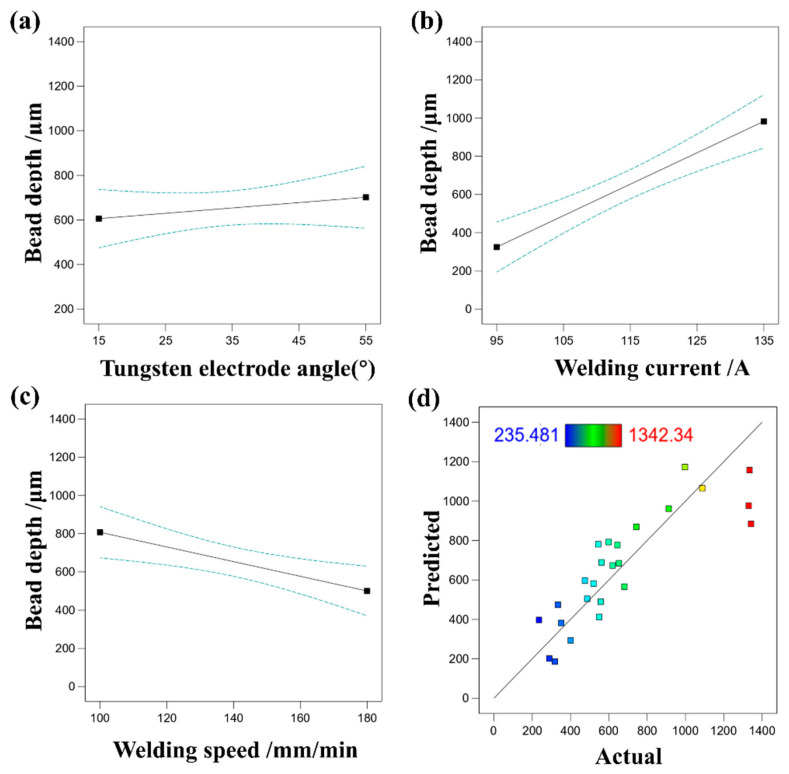
(**a**) The effect of tungsten electrode angle on bead depth, (**b**) the effect of welding current on bead depth, (**c**) the effect of welding speed on bead depth, (**d**) comparison of predicted and actual values of bead depth.

**Figure 9 materials-17-01337-f009:**
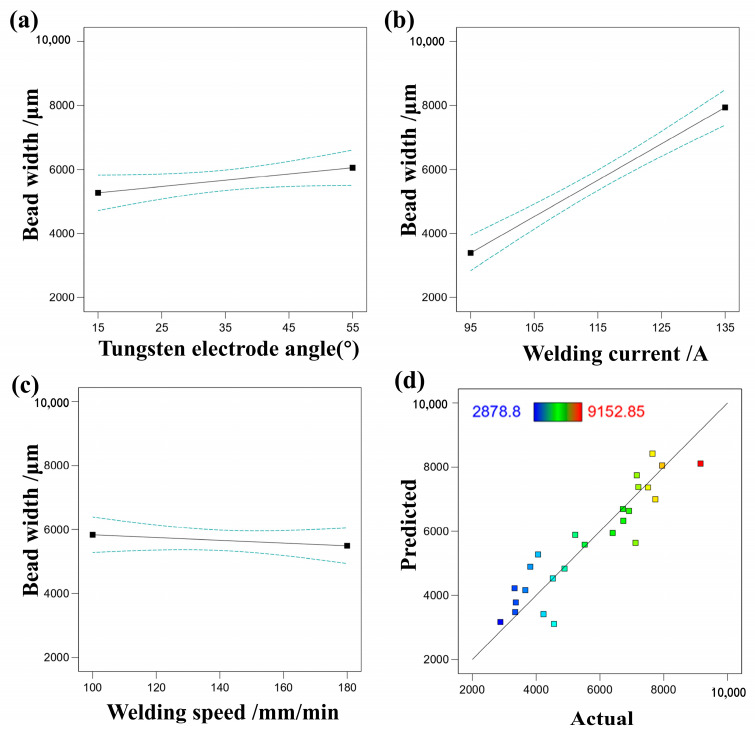
(**a**) The effect of tungsten electrode angle on bead width, (**b**) the effect of welding current on bead width, (**c**) the effect of welding speed on bead width, (**d**) comparison of predicted and actual values of bead width.

**Figure 10 materials-17-01337-f010:**
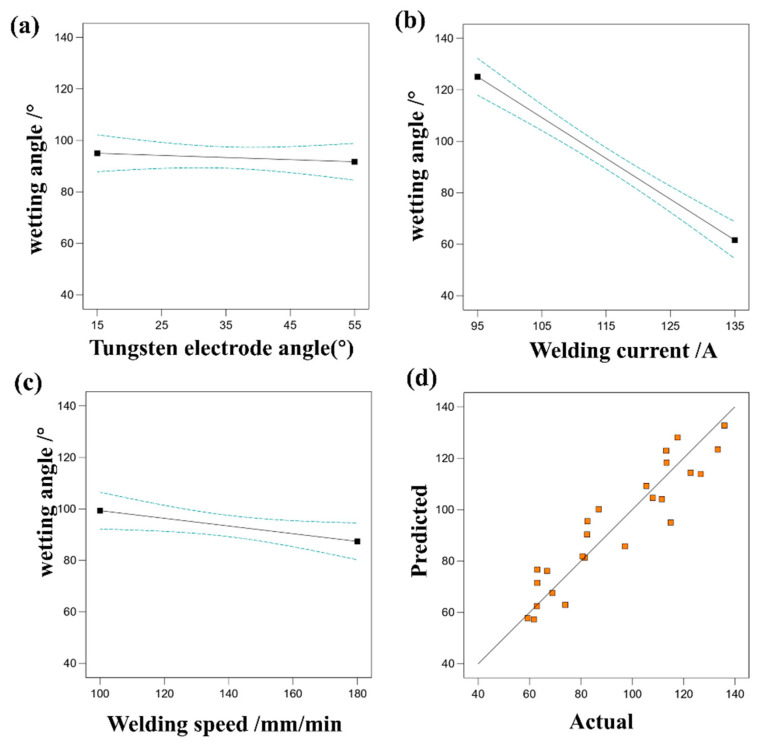
(**a**) The effect of tungsten electrode angle on wetting angle, (**b**) the effect of welding current on wetting angle, (**c**) the effect of welding speed on wetting angle, (**d**) comparison of predicted and actual values of wetting angle.

**Figure 11 materials-17-01337-f011:**
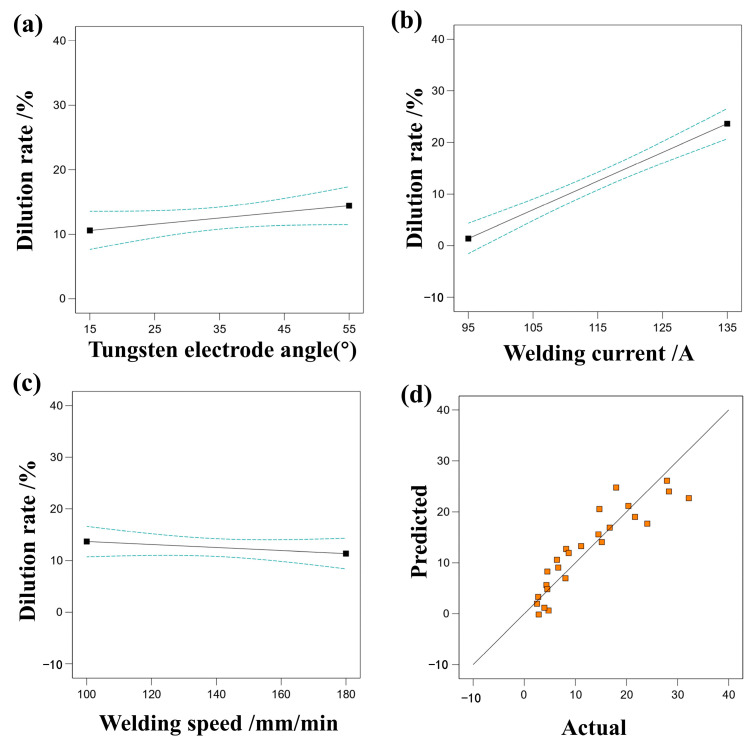
(**a**) The effect of tungsten electrode angle on dilution rate, (**b**) the effect of welding current on dilution rate, (**c**) the effect of welding speed on dilution rate, (**d**) comparison of predicted and actual values of dilution rate.

**Figure 12 materials-17-01337-f012:**
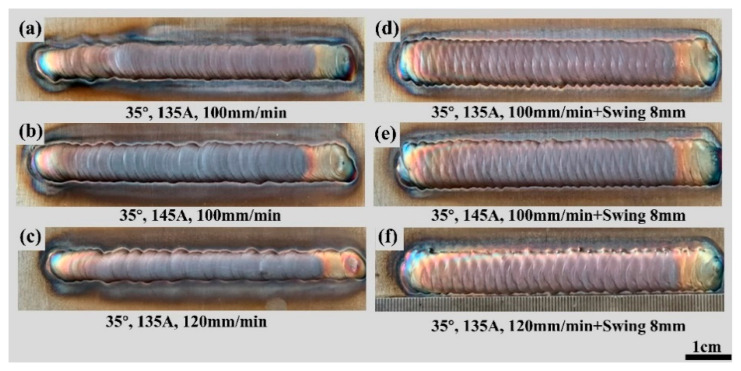
Shape of single-pass single-layer welds after parameter optimization and 8 mm swing: (**a**) 35°/135 A/100 mm/min, (**b**) 35°/145 A/100 mm/min, (**c**) 35°/135 A/120 mm/min, (**d**) 35°/135 A/100 mm/min + 8 mm swing, (**e**) 35°/145 A/100 mm/min + 8 mm swing, (**f**) 35°/135 A/120 mm/min + 8 mm swing.

**Figure 13 materials-17-01337-f013:**
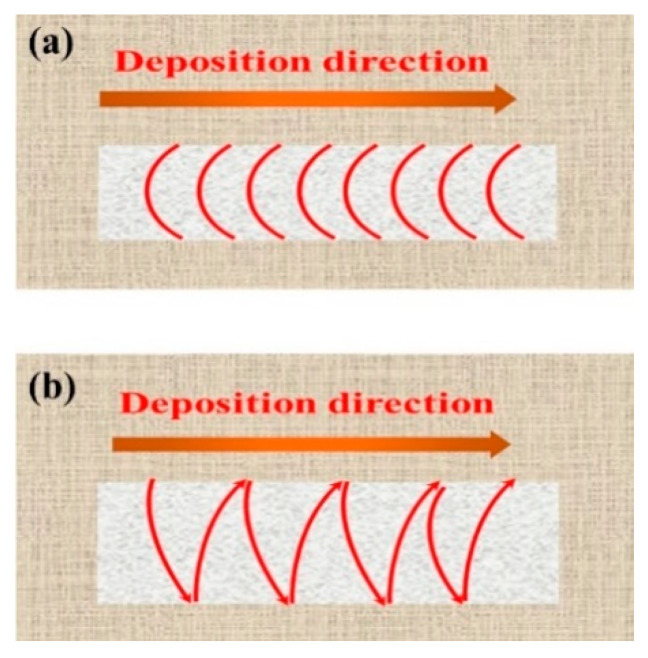
(**a**,**b**) are schematic diagrams of fish scale patterns formed by non-swing process and swing process, respectively.

**Figure 14 materials-17-01337-f014:**
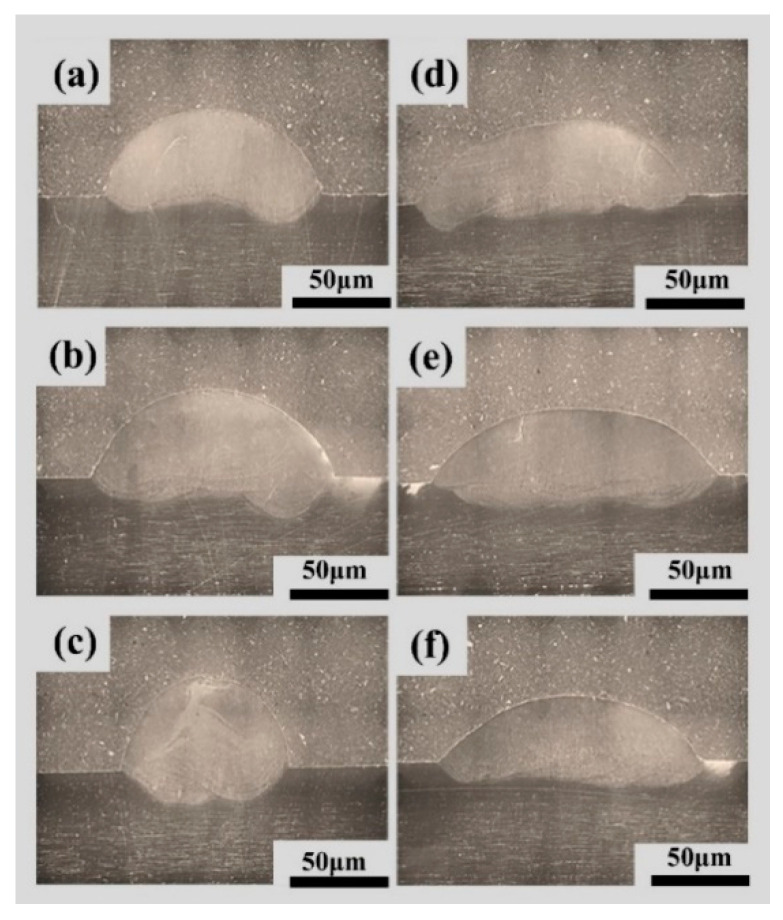
Cross-sectional view of the weld after parameter optimization and swing, (**a**) 35°/135 A/100 mm/min, (**b**) 35°/145 A/100 mm/min, (**c**) 35°/135 A/120 mm/min, (**d**) 35°/135 A/100 mm/min + 8 mm swing, (**e**) 35°/145 A/100 mm/min + 8 mm swing, (**f**) 35°/135 A/120 mm/min + 8 mm swing.

**Figure 15 materials-17-01337-f015:**
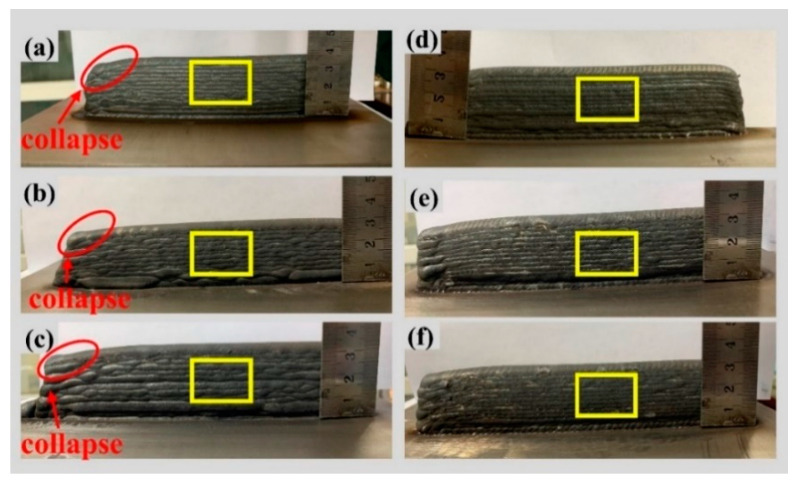
Macro morphology of single-pass multilayer wall forming after parameter optimization and swing, (**a**) 35°/135 A/100 mm/min, (**b**) 35°/145 A/100 mm/min, (**c**) 35°/135 A/120 mm/min, (**d**) 35°/135 A/100 mm/min + 8 mm swing, (**e**) 35°/145 A/100 mm/min + 8 mm swing, (**f**) 35°/135 A/120 mm/min + 8 mm swing.

**Figure 16 materials-17-01337-f016:**
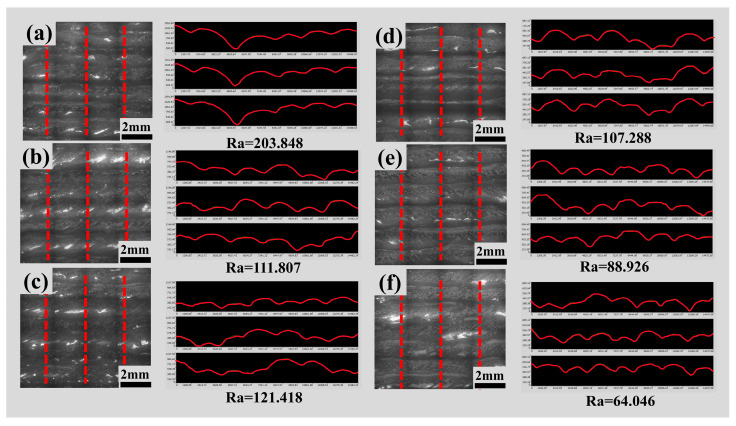
Local line roughness of single-pass multilayer walls after parameter optimization and swing, (**a**) 35°/135 A/100 mm/min, (**b**) 35°/145 A/100 mm/min, (**c**) 35°/135 A/120 mm/min, (**d**) 35°/135 A/100 mm/min + 8 mm swing, (**e**) 35°/145 A/100 mm/min + 8 mm swing, (**f**) 35°/135 A/120 mm/min + 8 mm swing.

**Figure 17 materials-17-01337-f017:**
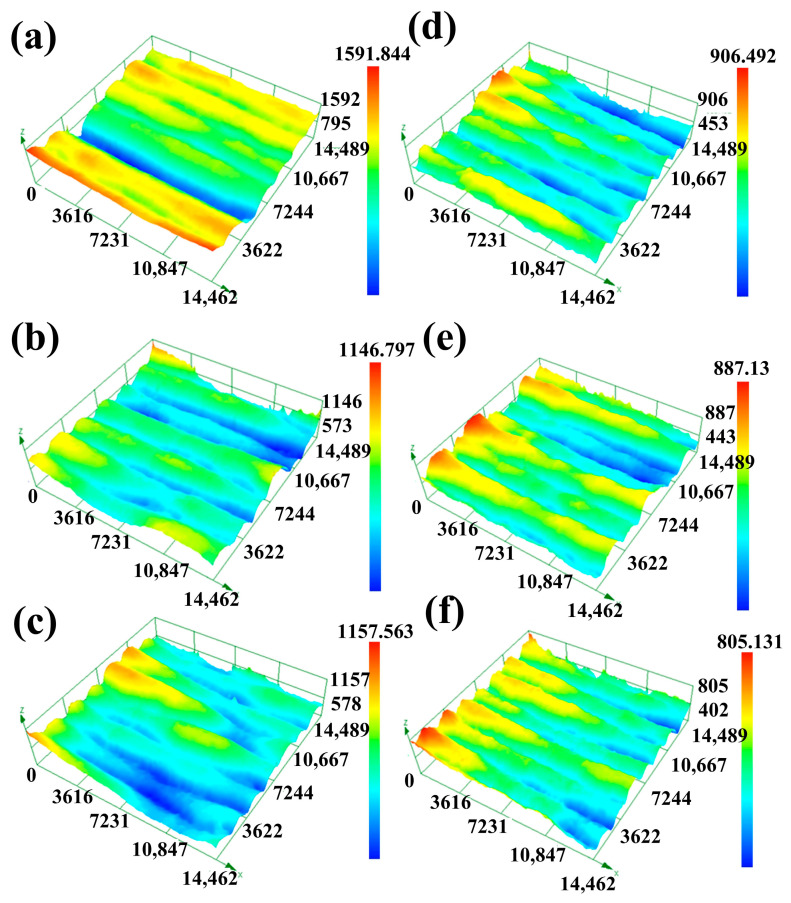
Local surface morphology of single-pass multilayer walls after parameter optimization and swing, (**a**) 35°/135 A/100 mm/min, (**b**) 35°/145 A/100 mm/min, (**c**) 35°/135 A/120 mm/min, (**d**) 35°/135 A/100 mm/min + 8 mm swing, (**e**) 35°/145 A/100 mm/min + 8 mm swing, (**f**) 35°/135 A/120 mm/min + 8 mm swing.

**Figure 18 materials-17-01337-f018:**
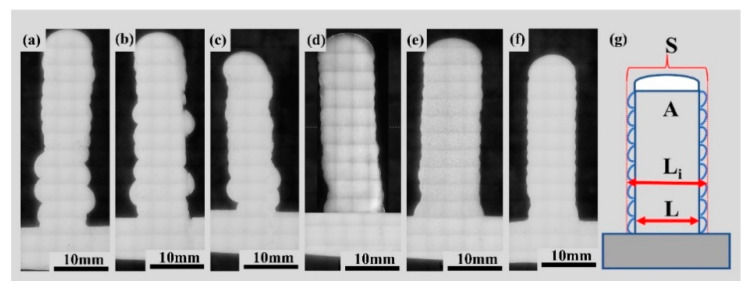
Single-pass multi-layer wall sections after parameter optimization and swing; (**a**) 35°/135 A/100 mm/min, (**b**) 35°/145 A/100 mm/min, (**c**) 35°/135 A/120 mm/min, (**d**) 35°/135 A/100 mm/min + 8 mm swing, (**e**) 35°/145 A/100 mm/min + 8 mm swing, (**f**) 35°/135 A/120 mm/min + 8 mm swing, (**g**) schematic of process efficiency.

**Figure 19 materials-17-01337-f019:**
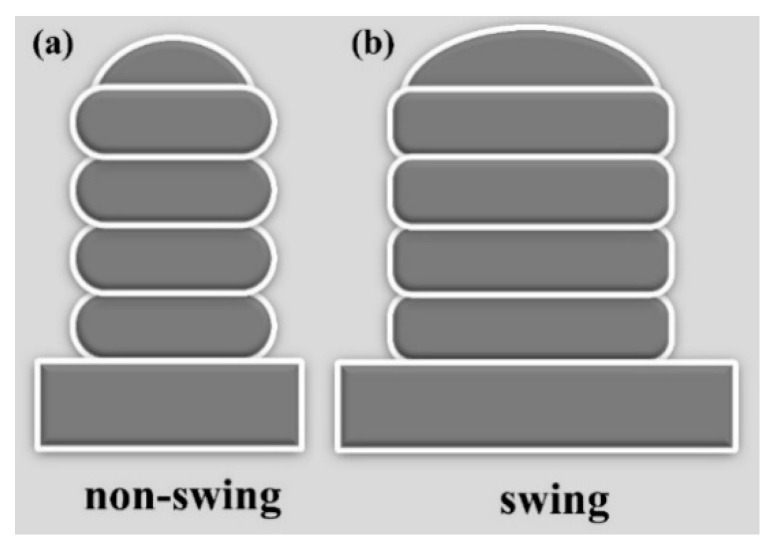
Schematic diagrams of non-swing and swing sections: (**a**) non-swing; (**b**) swing.

**Figure 20 materials-17-01337-f020:**
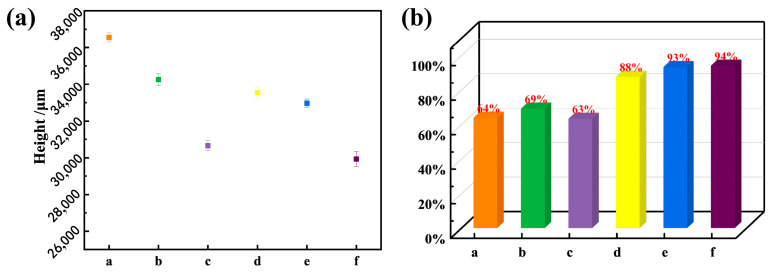
Height of single-pass multilayer wall after parameter optimization and swing (**a**) and forming efficiency (**b**), the figure caption (a–f) are respectively: (a) 35°/135 A/100 mm/min, (b) 35°/145 A/100 mm/min, (c) 35°/135 A/120 mm/min, (d) 35°/135 A/100 mm/min + 8 mm swing, (e) 35°/145 A/100 mm/min + 8 mm swing, (f) 35°/135 A/120 mm/min + 8 mm swing.

**Table 1 materials-17-01337-t001:** Chemical compositions of source materials (wt.%).

Materials	C	Mn	Si	S	P	Ni	Cr	Mo	Cu	N	Fe
304 substrate	0.05	1.09	0.45	0.003	0.032	8.00	18.08	0.012	0.14	0.054	Bal.
308L	0.022	2.12	0.515	0.006	0.023	9.75	19.92	0.032	0.031	0	Bal.

**Table 2 materials-17-01337-t002:** The process parameters that remain constant during the WAAM manufacturing process.

Process Parameters	Details
Distance of electrode to substrate	4 mm
Angle of nozzle to substrate	30°
Angle of nozzle to tungsten	60°
Gas flow rate of GTAW torch	15 L/min
Gas flow rate of trailing shield cover	15 L/min
Post-argon flow duration	5 s
Dwell time between layers	120 s

**Table 3 materials-17-01337-t003:** Variable process parameters.

Serial No.	Input Parameters	Symbols	Units	Levels				
1	Tungsten electrode angle	A	°	15	25	35	45	55
2	Welding current	B	A	95	105	115	125	135
3	Welding speed	C	mm/min	100	120	140	160	180

**Table 4 materials-17-01337-t004:** CCD design scheme and the result of the response value.

Serial No.	A: Factor 1	B: Factor 2	C: Factor 3	Response 1	Response 2	Response 3	Response 4	Response 5
	Tungsten Electrode Angle	Welding Current	Welding Speed	Bead Height	Bead Depth	Bead Width	Wetting Angle	Dilution Rate
	(°)	(A)	(mm/min)	(μm)	(μm)	(μm)	(°)	(%)
1	15	95	100	4617.848	549.335	2878.795	135.924	4.755%
2	15	105	120	4274.767	487.330	3324.733	126.695	4.338%
3	15	115	140	3633.638	474.838	4058.800	114.996	6.390%
4	15	125	160	2717.728	561.719	6731.990	66.843	14.511%
5	15	135	180	2414.458	544.756	7202.880	61.706	14.699%
6	25	95	160	4583.701	289.041	4558.627	113.251	2.829%
7	25	105	180	3614.209	400.788	3659.849	111.559	4.511%
8	25	115	100	3598.103	599.160	7114.498	86.976	8.197%
9	25	125	120	3027.317	1342.338	6719.678	81.501	24.081%
10	25	135	140	2448.061	1330.182	7153.620	62.860	32.191%
11	35	95	120	4296.385	235.481	3342.230	117.668	2.496%
12	35	105	140	3564.160	557.588	4522.106	105.016	8.057%
13	35	115	160	3010.714	520.264	5521.721	82.435	8.688%
14	35	125	180	2411.061	619.916	6910.416	63.019	16.717%
15	35	135	100	3066.646	997.576	9152.847	68.903	17.977%
16	45	95	180	4267.049	318.010	4230.122	113.386	3.938%
17	45	105	100	4460.113	652.508	3815.834	122.753	6.653%
18	45	115	120	3140.439	644.889	6398.481	82.586	15.170%
19	45	125	140	2511.521	743.472	7736.503	63.033	21.663%
20	45	135	160	2262.966	912.661	7948.921	59.236	28.306%
21	55	95	140	4890.840	351.168	3362.547	133.353	2.728%
22	55	105	160	4246.538	334.481	4888.712	108.007	4.517%
23	55	115	180	3626.270	681.448	5224.573	97.183	11.127%
24	55	125	100	3564.171	1087.386	7506.277	80.660	20.368%
25	55	135	120	3168.731	1334.422	7647.319	73.923	27.924%

**Table 5 materials-17-01337-t005:** ANOVA table for height of bead.

Source	Sum of Squares	df	Mean Square	F-Value	*p*-Value	Prob > F
Model	1.459 × 10^7^	9	1.621 × 10^6^	39.75	<0.0001	significant
A-A	47,639.54	1	47,639.54	1.17	0.2969	
B-B	1.047 × 10^7^	1	1.047 × 10^7^	256.66	<0.0001	
C-C	9.623 × 10^5^	1	9.623 × 10^5^	23.59	0.0002	
AB	21,543.47	1	21,543.47	0.5282	0.4786	
AC	0.0567	1	0.0567	1.389 × 10^−6^	0.9991	
BC	2.592 × 10^5^	1	2.592 × 10^5^	6.36	0.0235	
A^2^	1.002 × 10^6^	1	1.002 × 10^6^	24.56	0.0002	
B^2^	1.868 × 10^5^	1	1.868 × 10^5^	4.58	0.0492	
C^2^	91,353.02	1	91,353.02	2.24	0.1552	
Residual	6.118 × 10^5^	15	40,786.38			
Cor Total	1.520 × 10^7^	24				

R^2^—0.9598, Adjusted R^2^—0.9356, Predicted R^2^—0.8559, Adeq Precision—20.9835.

**Table 6 materials-17-01337-t006:** ANOVA table for depth of bead.

Source	Sum of Squares	df	Mean Square	F-Value	*p*-Value	Prob > F
Model	1.891 × 10^6^	3	6.303 × 10^5^	19.33	<0.0001	significant
A-A	54,574.48	1	54,574.48	1.67	0.2098	
B-B	1.505 × 10^6^	1	1.505 × 10^6^	46.17	<0.0001	
C-C	3.310 × 10^5^	1	3.310 × 10^5^	10.15	0.0044	
Residual	6.847 × 10^5^	21	32,602.45			
Cor Total	2.575 × 10^6^	24				

R^2^—0.7342, Adjusted R^2^—0.6962, Predicted R^2^—0.6260, Adeq Precision—13.6581.

**Table 7 materials-17-01337-t007:** ANOVA table for width of bead.

Source	Sum of Squares	df	Mean Square	F-Value	*p*-Value	Prob > F
Model	6.694 × 10^7^	3	2.231 × 10^7^	37.68	<0.0001	significant
A-A	1.916 × 10^6^	1	1.916 × 10^6^	3.24	0.0864	
B-B	6.466 × 10^7^	1	6.466 × 10^7^	109.19	<0.0001	
C-C	3.635 × 10^5^	1	3.635 × 10^5^	0.6139	0.4421	
Residual	1.244 × 10^7^	21	5.922 × 10^5^			
Cor Total	7.938 × 10^7^	24				

R^2^—0.8433, Adjusted R^2^—0.8210, Predicted R^2^—0.7772, Adeq Precision—17.2398.

**Table 8 materials-17-01337-t008:** ANOVA table for wetting angle of bead.

Source	Sum of Squares	df	Mean Square	F-Value	*p*-Value	Prob > F
Model	13,069.07	3	4356.36	44.08	<0.0001	significant
A-A	34.00	1	34.00	0.3440	0.5638	
B-B	12,589.10	1	12,589.10	127.39	<0.0001	
C-C	445.97	1	445.97	4.51	0.0457	
Residual	2075.23	21	98.82			
Cor Total	15,144.30	24				

R^2^—0.8630, Adjusted R^2^—0.8434, Predicted R^2^—0.8122, Adeq Precision—18.9664.

**Table 9 materials-17-01337-t009:** ANOVA table for dilution rate of bead.

Source	Sum of Squares	df	Mean Square	F-Value	*p*-Value	Prob > F
Model	1608.07	3	536.02	31.80	<0.0001	significant
A-A	45.82	1	45.82	2.72	0.1141	
B-B	1545.35	1	1545.35	91.68	<0.0001	
C-C	16.90	1	16.90	1.00	0.3280	
Residual	353.98	21	16.86			
Cor Total	1962.05	24				

R^2^—0.8196, Adjusted R^2^—0.7938, Predicted R^2^—0.7372, Adeq Precision—15.9977.

**Table 10 materials-17-01337-t010:** Parameter optimization and swing design scheme and the result of the response value.

	Bead Height (μm)	Bead Depth (μm)	Bead Width (μm)	Wetting Angle (°)	Dilution Rate (%)
a	3018.700	1051.654	9231.270	72.66°	2.528%
b	2786.026	935.615	11,650.265	54.957°	3.816%
c	3118.908	1520.438	10,409.224	59.894°	4.603%
d	2474.025	1129.722	12,233.838	51.758°	5.516%
e	3350.619	1227.085	7187.254	78.733°	3.386%
f	2376.039	1032.679	11,256.612	43.168°	4.712%

**Table 11 materials-17-01337-t011:** Forming height and forming efficiency.

	Number of Layers	Height (mm)	L (μm)	L_i_ (μm)	Forming Efficiency (%)
a	15	36,559.089	7241.048	11,328.519	63.921%
b	15	34,265.968	8016.998	11,591.895	69.157%
c	15	29,659.417	6656.058	10,509.323	63.336%
d	15	33,539.914	9501.562	10,842.430	87.631%
e	15	32,969.751	9628.981	10,329.996	93.214%
f	15	30,933.713	8404.616	8939.096	94.015%

## Data Availability

Data are contained within the article.

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
