# Peer review of "The Effects of Processing Parameters during the Wire Arc Additive Manufacturing of 308L Stainless Steel on the Formation of a Thin-Walled Structure"

_materials, 2024, doi:10.3390/ma17061337_

Round 1

Reviewer 1 Report

Comments and Suggestions for Authors

This work studies effects of processing parameters during wire arc additive manufacturing of 308L stainless steel on the formation of thin-walled structure. The research is well-designed and presented clearly. A good comparative analysis of existing publications and the tasks set in the work is carried out. The methodological section of the manuscript is presented in sufficient detail. The authors used modern equipment for the preparation and testing of samples. They also utilized the equipment for visualization and assistance in interpreting the obtained results. According to the authors’ findings, the model analysis and experimental results within the study indicate that welding current exerts the most significant influence on weld seam formation, while welding speed and tungsten electrode angle have relatively minor impacts. Additionally, there is a notable interaction effect between welding current and tungsten electrode angle on weld seam height. The authors found the optimal parameters to form thin-wall structures to be as follows: welding current of 135 A, welding speed of 120 mm/min, tungsten electrode ange of 35°, and a swing of 8 mm.

However, some shortcomings should be corrected to make the manuscript acceptable for publication in Materials.

(1) The title of the manuscript should be corrected. It should be either “… on the formation of thin-walled structure” or  “… on forming thin-walled structure”.

(2) The text in Figures 1, 2, 17, and 20 cannot be recognized.

(3) In Figures 6(a), 8(a), 9(a), 10(a), and 11(a), the name of the horizontal axis should be corrected, namely, “Tungsten angle (°)” should be replaced with “Tungsten electrode angle (°)” or “Electrode angle (°)”. The same concerns the captions of Figures 6, 8, 9, 10, and 11: Instead of “The effect of angle of tungsten electrode angle, …”, it should be “The effects of tungsten electrode angle, …” The same concerns the main text (Lines 114, 143, 279), Table 3, Table 4, the Conclusions section (Line 502), the caption of Figure 7, and the name of the horizontal axis in Figures 7(a) and 7(b).

(4) Is the parameter Li in Figure 18 the same as the parameter L1 in Table 11?

Comments on the Quality of English Language

In my opinion, the English language of this manuscript should be slightly improved.

Reviewer 2 Report

Comments and Suggestions for Authors

Dear authors

I have overall enjoyed article Reading. The topic discussed by the authors is interesting to the readers and in general the article is well written. I list below some major and minor and changes that must be addressed before further article processing.

Major corrections

I recognize the effort placed on the multiple experimental procedures that yielded vast amount of data. With this information, the authors could develop empirical models for the influence of processing parameters on the overall characteristics of additive manufacturing. What is missing in the manuscript is a comparison between the developed empirical models and previous theoretical studies i.e. are the empirical models from this manuscript consistent with already published studies? If not, what are the main differences?

Following with previous concern, the authors provided in the conclusion section a detailed set of parameters for optimal results in additive manufacturing using welding. Are these results consistent with previous studies within the field?

Minor corrections

Line 72: This statement requires some citations, i.e. Please cite previous theoretical models.

Figures 2 and 17: Axes of this figure are barely readable, please consider image resizing. The legend of Figure 2(f) is not readable either.

Comments on the Quality of English Language

Line 65: Considering that previous work from Sun et. Al. was reported, the verb focus should be used in past tense.

Reviewer 3 Report

Comments and Suggestions for Authors

The paper entitled “Effects of processing parameters during wire arc additive manufacturing of 308L stainless steel on forming of thin-walled structure”  deals with studies on processing parameters during the wire arc additive manufactured 308L stainless steel manufacturing with orthogonal experiments design and response surface methodology.

The work was planned correctly and resulted in valuable results. The methods were selected correctly to get the answers to the questions related to the topic. The design matrix was selected based on three process parameters with five levels of variables. The results are discussed in detail and correctly. The authors found a notable relationship between the welding current/tungsten electrode angle and weld seam height. The optimal parameters were determined. Thus, I can suggest accepting it to publish after some minor revisions.

1., For example,  “angle” is twice in one sentence (lines 70,71), one among them should be deleted.

2., Line 121, The metallographic preparation part should be written a bit more clearly, e.g. the “corrosion” is not a correct expression for the process (etching much closer to the real processes). It is not a question, that aqua regia can corrode the stainless-steel surface due to its chloride content (pinholes are formed, but only during a longer time.  Nitric acid can oxidize the stainless-steel surface with the formation of an oxide layer, and the HCl content of aqua regia can dissolve that. So the surface is cleaned and due to short contact time is passivated.

Round 2

Reviewer 2 Report

Comments and Suggestions for Authors

Mandatory changes have been replied one by one. It can be published in present form.